# Emotion Elicitation Under Audiovisual Stimuli Reception: Should Artificial Intelligence Consider the Gender Perspective?

**DOI:** 10.3390/ijerph17228534

**Published:** 2020-11-17

**Authors:** Marian Blanco-Ruiz, Clara Sainz-de-Baranda, Laura Gutiérrez-Martín, Elena Romero-Perales, Celia López-Ongil

**Affiliations:** 1University Institute on Gender Studies, Universidad Carlos III de Madrid, 28903 Getafe, Spain; mangeles.blanco@urjc.es (M.B.-R.); lagutier@ing.uc3m.es (L.G.-M.); Spaineleromer@ing.uc3m.es (E.R.-P.); celia@ing.uc3m.es (C.L.-O.); 2Department of Communication Sciences and Sociology, Faculty of Communication Sciences, Universidad Rey Juan Carlos, 28943 Fuenlabrada, Spain; 3Department of Communication and Media Studies, Faculty of Humanities, Communication and Library and Science, Universidad Carlos III de Madrid, Getafe, 28903 Madrid, Spain; 4Electronic Technology Department, School of Engineering. Universidad Carlos III de Madrid, Leganés, 28911 Madrid, Spain

**Keywords:** emotion elicitation, audiovisual stimuli, gender, pleasure-arousal-dominance affective space (PAD space), machine learning, UC3M4Safety database

## Abstract

Identification of emotions triggered by different sourced stimuli can be applied to automatic systems that help, relieve or protect vulnerable groups of population. The selection of the best stimuli allows to train these artificial intelligence-based systems in a more efficient and precise manner in order to discern different risky situations, characterized either by panic or fear emotions, in a clear and accurate way. The presented research study has produced a dataset of audiovisual stimuli (UC3M4Safety database) that triggers a complete range of emotions, with a high level of agreement and with a discrete emotional categorization, as well as quantitative categorization in the Pleasure-Arousal-Dominance Affective space. This database is adequate for the machine learning algorithms contained in these automatic systems. Furthermore, this work analyses the effects of gender in the emotion elicitation under audiovisual stimuli, which can help to better design the final solution. Particularly, the focus is set on emotional responses to audiovisual stimuli reproducing situations experienced by women, such as gender-based violence. A statistical study of gender differences in emotional response was carried out on 1332 participants (811 women and 521 men). The average responses per video is around 84 (SD = 22). Data analysis was carried out with RStudio^®^.

## 1. Introduction

### 1.1. Human Emotions and Their Categorization

Identifying the emotion experienced in a given situation, or the emotion elicited when visualizing audiovisual stimuli, can be very interesting in several applications intended to improve people’s life (human-machine interfaces, mental health, industrial design, neuro-marketing, etc.). Emotion is defined as a psychological state including three components: subjective personal experience, associated physiological response and behaviours [1]. In the different studies on the basic emotions, these have been categorized as discrete concepts that can be identified and classified. These studies propose that human beings have a limited number of emotions, each expressed in an organized recurring pattern of behavioural elements [2,3,4]. The first list of basic emotions made by Ekman was composed of anger, fear, surprise, sadness, disgust and happiness [4]. Since then, larger lists and other types of representation have been proposed [5,6,7,8,9]. There have been some research works that have analysed the discrete representation of emotions. Many of them concluded there is an important socio-cultural factor in emotion discrete values understanding or, even, there is a lack of completeness in the emotion discrete values, biasing them to negative emotions due to the large amount of mental health studies [10,11,12]. These inconveniences have been partially solved enhancing and balancing the discrete emotion values list, but additional information about the level of intensity in the emotion felt is still lacking. This intensity information can be valuable information for an emotion classifier and its decision-making behaviour for artificial intelligence-based systems.

More recently, multidimensional numerical values have been proposed to represent emotions in the Affective Space, which allow a more accurate emotion representation and, therefore, a better operation of systems requiring emotion classification. For example, Fontaine et al. [13] propose an Affective Space with three dimensions (valence, arousal and dominance). Valence ranges positive to negative feelings, from disgust to pleasure. Arousal ranges from calm to excitement states. Finally, dominance corresponds with the control capability over the emotion intensity [14]. These variables allow locating the emotions in a three-dimensional space (Pleasure, Arousal, Dominance-PAD) so that every emotion is in a unique space, is not overlapping the others. Generally, authors work with the bi-dimensional space PA that divide emotions in positive and negative (Pleasure-Valence), as well as in very activating or calming (Arousal). This makes four quadrants to locate emotions (hereafter referred to as Q1, Q2, Q3 and Q4). Some research works locate emotions in these four quadrants, although some of them are overlapping. Only when the third dimension, dominance, is used, it is possible to separate the overlapped emotions [15,16]. For example, fear and anger, in quadrant Q2, or surprise and joy in Q1. Figure 1 presents an outline of this bi-dimensional PA space we worked with in this research, showing a tentative position of some basic emotions.

### 1.2. Triggering Emotions and Its Usefulness

The identification of what emotions are triggered by different sourced stimuli can be applied to automatic systems to help, relieve or protect some groups of population. Both elements, stimuli and labelled emotions are the primary inputs for an artificial intelligence-based system; that, of course, needs from secondary inputs, other types of human responses (the more unconscious the better) such as physiological variables, voice, face expressions, etc. to train the algorithms that could classify, in real time and automatically, the emotion experienced by the person. In this sense, Bindi system is proposed by the UC3M4Safety team to detect and prevent violent aggressions against women, by detecting fear or panic emotions, through the voice and the physiological responses [17,18,19]. However, specific databases are required to create these artificial intelligence-based algorithms in detecting fear or panic emotions and considering the users will be women. The selection of the best stimuli allows us to train and test these automatic systems in an efficient and accurate way. Although real life experiences are the best way to trigger emotions at its highest intensity, this is not always possible and even worse, repeatable for artificial intelligence systems training. Therefore, there are scientific audiovisual databases specially aimed for emotion studies. For example, FilmStim [20], MANHOB [21], DEAP [22] or Emotional Film Database for Asian Culture [23] are largely used and referenced. They also use the methodology of Self-Assessment Manikin (SAM) from Lang [24] as the procedure for emotion labelling in the PAD space, with an acceptable global reliability [3].

These databases use as stimuli audiovisual resources from popular culture (e.g., films, music videos, etc.) to trigger many of the emotions that are experienced daily, as Ellis defines “the reality effect of moving images and sounds invokes many of the emotions that we experience during direct encounters in our lived spaces” [25]. In fact, an audiovisual clip is able to induce an emotional state.

The identification with the characters [26] is a central concept to understand and to explain the emotional processes and effects of the audiovisual fictional works in the audience. Through audiovisual the autobiographical memory activation is produced. This cognitive process-also known as emotional memory, referential reflection or personal resonance-is one of the main processes happening between the audience and the story, and its intensity depends directly on the implication degree with the story.

Personal resonance resulting from audiovisual stimuli triggers emotions that, being related to personal events, activate autobiographical memories. The key for these processes of assessing emotions elicited by audiovisual stimuli is the empathy with the characters or account. The empathy is the social ability to identify with the other one and share his/her feelings [27,28] (for a general view of definitions). Departing from the observation of the other one, projecting her/himself in us, internalizing and setting an interaction through we can acquire the capability of suffering, cheering, and, definitely, feeling with the other one [29].

According to authors such as Cohen [26] or Igartua [30,31], empathy with audiovisual stories can be divided in four different forms of identification with the characters: empathic concern that refers to the ability to feel what characters are feeling, affectively involving oneself or feeling worried by their problems and experimenting social-emotions (“feel with/for the characters”); cognitive empathy which refers to understand or to put oneself in the characters’ shoes, which is related with the ability of taking the other one’s perspective or point of view (perspective taking); the experience of becoming the character and the loss of self-awareness is referred to the sensation of feeling as if oneself were one of the characters; and, finally, the personal attraction towards the characters which is linked with the subprocesses of positive appreciation of characters, the perception of similarities with them and the desire of being one of them.

### 1.3. Research Objectives

When comparing the different emotion recognition systems from the literature, which can be subject-dependent or subject independent, the personalization problem arises. Most of the differences between participants lie in the dynamic nature of their affective states and their previous experience. Hence, stimuli interpretation is strongly volunteer-dependent [12]. One of the most studied cases is the emotional response of people suffering from post-traumatic stress as a result of a traumatic experience such as war [32,33,34,35], terrorist attack [36] or natural disaster [37,38]. Fears associated with traumatic experiences are not common to all people [39] and change their perception of risk.

Different experiments performed for emotion recognition have concluded that, apart from individual differences [40,41], there are cultural, language, gender and age differences [42,43,44,45,46] that should be considered and adapted to achieve a better empathizing with those audiovisual stimuli under visualization. Besides, there are scientific studies that have concluded women recognize more accurately the nonverbal communication or emotional prosody [47] and are more sensitive to emotional expressions in interpersonal interactions [48]. Regarding physiological variables, even though women reported lower values of valence and higher values of arousal than men for unpleasant films gender differences were not found in skin conductance, heart rate or electromyography (EMG) activity when men and women are exposed to high-arousal’s level pleasant and unpleasant films [49].

Particularly for fear, women tend to report more intense fear when watching films where the victim is also a women [50]. Subsequent works found gender differences in electrocardiogram (ECG) signals (especially for non-linear characteristics) when exposing men and women to sadness stimuli [51]. These results are not currently considered in any emotion recognition system using physical or physiological signals presented in the literature.

From all these research works arises the complexity of building an adequate set of audiovisual stimuli that could provoke emotions in a realistic, repeatable, and accurate way. The presented research study analyses the effects of gender in the emotion elicitation under audiovisual stimuli to assess if this perspective should be considered in the data set building, aimed to classify human emotions in artificial intelligence-based systems. Our initial hypothesis is that men and women have different emotional responses, due to gender socialization [52,53]. Differences in emotional response that may be increased for women who have experienced gender-based violence because post-traumatic stress and risk perception is analogous to the aftermath of war victims [54]. In particular, the focus is set on emotional responses to audiovisual stimuli reproducing situations experienced particularly by women, such as gender violence.

The intended set of stimuli is purposed to build an audiovisual database for emotion identification with artificial intelligence. This database is called UC3M4Safety and its aim is to train different machine learning algorithms in further work within the research group related to vulnerable population group’s protection. As mentioned, Bindi system is aimed to protect gender-based violence victims. For this purpose, the composition of UC3M4Safety Database aims to train devices in the identification of dangerous situations (see Section 2.1 Materials).

To summarize, the research questions leading this work are:(1)Are emotions best labelled in the PAD space or with the discrete values?(2)Is there a high level of agreement in the reported emotions?(3)Are emotional responses different for women and men?(4)Can we select a representative and small set of audiovisual stimuli to elicit emotions adequately?

## 2. Materials and Methods

### 2.1. Materials

In this research work the materials used are audiovisual clips (that make up the UC3M4Safety audiovisual database) whose aim is to trigger different emotions in persons. In the creation of the UC3M4Safety audiovisual database, the clips are originally labelled with a target emotion by the research team, with the advice of a panel of experts, and shown to a large set of volunteers who label again every clip with the experimented emotion after stimuli visualization. The reported emotions are not always equal to the previously labelled. The analysis of these reported emotions will serve for a double purpose: the selection of a set of audiovisual stimuli for training smart autonomous systems that detect fear in vulnerable people and the observation of differences in emotion elicitation from the gender perspective.

The selection of the set of audiovisual stimuli (UC3M4Safety database) has been performed in five steps, as shown in Figure 2. Every step refined the selected stimuli to assure the required quality for the research purposes. In Figure 2, every coloured box states for the video clips set output in the steps, with the criteria applied for the selection, while the white boxes note the process of refinement and the actors implied in the process, between brackets.

Firstly, over a period of four months, five researchers collected samples of films with emotional content from commercial films, series, documentaries, short films, commercial advertisements, and video clips from the Internet. From this larger pool, short film clips were created by editing key scenes using Camtasia^®^ (TechSmith Corporation, Okemos, MI, USA) and Adobe Premiere (Adobe, San José, CA, USA). The protocol followed to collect all the film clips was adopted from Gross and Levenson [55], Jenkins and Andrewes [56] and Deng et al. [23]. The discrete emotions contained in the audiovisual stimuli sought by the researchers were joy, sadness, surprise, contempt, hope, fear, attraction, disgust, tenderness, anger, calm and tedium. The list of emotions for this research (Table 1) was obtained from the coincidences in the studies by Ekman [4,5,6,7], Plutchnik [8], Robinson [1,12] and Izard [9], taking into account the variables used in other previous audiovisual bases such as FilmStim [20], MANHOB [21], DEAP [22], Emotional Film Database for Asian Culture [23], but incorporating Ekman’s recent contributions [57,58] and Robinson’s work [12], among others, that any emotion can be represented in a positive/constructive or negative/destructive way.

Secondly, from the 370 samples obtained in the first step, 162 videos were selected for further evaluation based on the criteria: (1) There were no visible watermarks, logos, or mosaics; (2) the thematic content was understandable without additional explanation; (3) the film elicited only one target emotion and not non-target emotions; (4) the level of excitement of the emotion had to be appropriately high; and (5) the length of the film clips had to be relatively short, maximum 2 min. In the case of gender-based violence clips, another criterion was added: the main actors in the films had to be women who suffered some kind of violence (sexual, physical, psychological, etc.).

The intention was to survey the emotion elicitation for this list of 162 video clips in a large number of volunteers to extract conclusions that help to build a set of audiovisual stimuli (UC3M4Safety database). Also, all the target emotions had to be contained in the clips, uniquely per video, and with a representative number of samples. Therefore, in the list, every emotion was represented by around 10 video clips, except for the fear emotion, which was the main objective in this research work, for which 40 clips were selected. The number of video clips per target emotion is reported in Table 2.

Thirdly, the list of 162 video clips was surveyed in an online poll (see the next section for the actual procedure) although different visualizations were produced. Only 80 video clips obtained enough answers to be considered for further analysis (more than 50 visualizations were the threshold). The number of video clips for the different target emotions complying with this threshold is also provided in Table 2, in parentheses, being the global number 80. For the final selection of the audiovisual stimuli (step 4, see Figure 2), two conditions were set: the first one looked for the highest percentages of agreement among the participants, meaning at least 50% of the volunteers considering both genders together or at least 50% of one gender individually, who visualized each stimulus labelled it with the same discrete emotion. At the same, the second imposed condition checked the uniqueness of that label, by ensuring that all the other possible emotions only reached as maximum a 30% agreement. Finally, due to the complexity shown by the “anger” emotion, which even worsened when the clip was related to gender-based violence, the research team decided to keep the stimuli labelled with this target emotion which reached at least 40% of agreement to be able to study the subsequent label evolution in future experiments with gender-based violence victims. Only in four videos (V08, V09, V10, V29) the objective emotion labelled by the researchers was changed by the majority of the volunteers. This reduced the original set of videoclips up to 53. However, as previously mentioned, due to the fact that in machine learning is recommended to train the algorithms with a balanced dataset, some videos were discarded to obtain an even distribution between the target emotions considered as “fear” and “not fear” (step 5, see Figure 2), producing a selection of 42 clips. the distribution among quadrants can also be observed in Table 3; although two emotions have no video clips in the final set, the global number of emotion per quadrant is around 6–7, except for Q2 (where fear emotion is included) with the same number of video clips as the other three quadrants together. Also, seven videos (V08, V09, V11, V19, V21, V27 and V28) were labelled by the research team as video clips with interest in their content related to gender-based violence.

Therefore, the final number of audiovisual stimuli complying with these two conditions, and the result of this research (denominated UC3M4Safety database) were 42 clips. The complete process of materials is detailed in Figure 2. More details are provided in the Results section, together with an analysis of the emotion labelling for the clips, with gender perspective. Further details of the films selected for the UC3M4Safety Database can be found in the Appendix A.

### 2.2. Procedure

The main task executed on the materials for this research was the survey with a large number of volunteers. The 162 selected film clips were organized in 42 online questionnaires that were distributed online by the snowball procedure [59]. Each questionnaire consisted of 4–6 videos of different emotions. The questionnaires were distributed between March and June 2020. Every film clip evaluated by at least 50 people was selected for further analysis. The experimental procedures and data protection methods were approved by the Carlos III University of Madrid Ethics Committee.

In the experiment we collected as data the subjective experience reported by volunteers. Each participant completed a questionnaire of 20 self-evaluation items about the emotion experienced during the clip viewing in relation to: valence, arousal, dominance, likability and familiarity, and finally, they chose the discrete emotion (12 items: joy, sadness, surprise, contempt, hope, fear, attraction, disgust, tenderness, anger, calm and tedium) that best described their emotional state during the film.

This rating procedure was similar to that used by previously related research works [14,20,22,23,55]. It was a 9-point Likert scale. For the familiarity and likability items the anchor points of 1 and 9 correspond with “not at all” and “very much,” respectively. Participants rated valence, arousal and dominance using the Self-Assessment Manikin [60]. For valence the anchor points were 1 for “very unhappy” and 9 for “very happy;” for arousal 1 for “very calm” and 9 for “very aroused”; and for dominance the rating was 1 for “feeling totally dominated” and 9 for “feeling strong control power”.

The trial order was designed so that: (1) no two films targeting the same emotion were shown in a row, (2) no more than two films of a particular valence (negative or positive) were shown consecutively and (3) each film had the same chance to be shown in every order for different participants.

Prior to watching the films, the volunteers signed a consent form and answered demographic questions. With regards to data protection, and following the current European law, the consent form informs about the usage of the collected data from the volunteer, as well as the possibility of future withdrawals from the data owner. The collected data were pseudo-anonymized for further analysis, assuring the no possible identification of volunteers and the capability of eliminating withdrawn data. Also, in the consent form, the research team stated that the purpose of the study was to research about emotions, specifically fear. Participants were told that the films would be shown on a screen and they should watch the film carefully, but could look away or shut their eyes if they found the films too distressing or could stop the experiment if they felt uncomfortable at any time.

### 2.3. Sample

The analysis units were the 42 videos of audiovisual stimuli selected for the UC3M4Safety database. A statistical study of gender differences in emotional response was carried out for the 1332 participants (811 women and 521 men) corresponding to the selected videos. The participants were 18–78 years old (mean age 38.27, SD = 14.47), and all were Spanish speakers. All of them had normal (or corrected) vision and auditory functions. A summary of these 42 video clips visualizations is presented in Table 4. The average responses per video is around 84, while the standard deviation (SD) is around 22. Regarding gender, the mean of responses per video clip from women is 46 while 37 from men, with SD of 14 and 11, respectively. In general, there have been more visualizations from women than from men. This fact can be associated with the more likeness of women to participate in public polls [61].

### 2.4. Data Analysis

Data analysis was carried out with RStudio^®^ (RStudio, Boston, MA, USA). These analyses have been performed on the 42 video clips resulting from step 5 (Figure 2). Firstly, a global analysis on the data collected was carried out, a confusion matrix was computed to visualize two important results: If the target emotion selected by the experts was the same to the reported emotion obtained from the volunteers and which videos could be included in the selection for a subsequent analysis as they surpassed the thresholds of discreteness and agreement set.

Once the 42 videos were selected, all the data belonging to the same video, even the observations with a reported emotion not matching with the target, were used to compute its multidimensional position using the median score for each of the parameters (valence, arousal and dominance) to compare the actual position associated to each video and the original one. We use the median to ensure the robustness of the data and avoid the presence of distribution biases or extreme values.

In addition, to observe tendencies of dependency related to gender analysis, and after checking the homogeneity of the variance (Levene’s test) and normality (QQ-plot) assumptions, repeated measures of ANOVA were performed for the PAD variables reported (valence, arousal and dominance) and the discrete emotion associated to it (joy, sadness, surprise, indifference, hope, fear, attraction, disgust, tenderness, anger, calm, tedium), for every emotion reported.

Finally, contingency tables for the bivariate analysis were used, based on the observed results, aimed at specifying and analysing the characteristics of the sampling used in the study. A non-parametric inferential technique was carried out (Hypothesis Test-$\chi^{2}$ Test of Independence) to study the possible relationships between the analysed variables. The significance level was set at *p*-value < 0.05. Additionally, and because our main variable (the reported emotion) has 12 levels, a post-hoc analysis was performed to find specifically where this relationship exists. The study of the standardized Pearson residuals, which tried to find differences between the observed and expected values.

## 3. Results

Once data was filtered to keep only the answers related to the 42 selected video clips, a complete process of data analysis was performed. Firstly, emotions reported are analysed for the two types of classification: discrete and numerical (PAD space) for all the participants (paying special attention to agreement of reported emotions and with a gender point of view). Secondly, a detailed analysis for every quadrant in Affective Space on the labelling emotions by women and men is presented.

### 3.1. Reported and Target Emotions: Discrete Emotions and PAD Space (Numerical Values)

Regarding global results and emotions reported with respect to target ones, in Figure 3 the multidimensional position for every video clip is displayed in the Affective Space (Arousal-Valence). In Figure 3, median values (reported) are represented for valence on the horizontal axis and arousal on the vertical axis, while the shape of the dots represents the target quadrant. Also, the target discrete emotion is shown with a colour scale, detailed in the figure. Specially, those video clips whose target emotion was located in quadrants 1 and 3 (circle and square dots) have obtained a median reported label displaced upwards and downwards respectively, being arousal the parameter not matching with the expected values. Besides, there is a difference in the arousal reported for negative emotions (5–9 in the Likert scale) and positive emotions (always lower than 6).

When comparing target emotion and target quadrant for every video and the reported position in the AV space from the same volunteers, there are a subset of videos with non-negligible divergences. The main problem arises with the intensity of experienced emotion (arousal) which is reported very low for positive emotions in Q1 (joy, hope, surprise and attraction) and very high for negative emotions in Q3 (sadness, contempt, tedium and disgust). For example, video clip V06 (surprise) was labelled by 65 volunteers, with 51% of agreement on the same target emotion. Regarding quadrants: 80% of volunteers labelled emotions belonging to quadrant Q1, 0% to Q2, 8% to Q3 and 12% to Q4; but the PA parameters reported locate the video clip in Q4 as reported arousal is below 5. Similarly, video clip V31 (disgust) was labelled by 66 volunteers, with 67% of agreement for the same target emotion. Regarding quadrants: 68% volunteers labelled emotions corresponding to quadrant Q3, 15% to Q1, 12% to Q2 and 6% to Q4; but the PAD parameters reported locate the video in Q2 as reported arousal is above 5.

Even, more differences are observed for the video clip position in the Affective Space regarding the gender of volunteers. Therefore, the possible dependence between these numerical parameters (arousal, valence and dominance), the gender and the discrete emotion reported has been studied and presented in the following paragraphs.

The reported values of valence, arousal and dominance (see Table 5), has been analysed and clear differences have been observed between men and women, confirmed after ANOVA test (see Table 6). The mean values and the standard deviations for the three parameters per emotion and per gender, as well as the global values, are given in Table 5.

In the reported valence parameters, where the anchor points were 1 for “very unhappy” and 9 for “very happy” women tend to report higher values than mean for joy, surprise, hope, attraction, disgust, tenderness, calm and tedium emotion video clips, matching better the objective value (high or low) in all emotions except in disgust and calm. On the other hand, men report higher mean values than mean for anger, contempt and sadness, matching better the objective value (high or low) in those emotions requiring less extreme valence values (disgust and calm). The larger differences between valence mean values reported by men and women are given in calm (0.5) and fear (0.5) for women who match closer to the extreme values (positive and negative respectively) and in contempt (0.4) for men. In all the emotions the differences between men and women are not very high for the mean values and low for the SD.

In the case of arousal parameter, where the scale was from 1 for “very calm” to 9 for “very aroused”, differences between men and women are higher than in the other two parameters, and both reported values in some emotions which are far from the expected (joy, sadness, surprise, hope, attraction and disgust) according to the literature (detailed in Section 1 Introduction). Women reported higher values in stimuli corresponding to anger, fear, contempt, sadness, tedium and tenderness emotions, matching better the expected values in anger, fear (high values) and calm (low values). In self-reported arousal by men, the discrete emotions of attraction, hope and calm score higher values than the mean, matching better the expected values in sadness, contempt, tenderness and tedium (low values) and in hope and attraction (high values). The larger differences between arousal values reported by men and women are given in contempt and hope (0.7) where women reported centred values; also there are differences around 0.4/0.5 in fear, attraction, sadness, calm and tedium being extreme expected values always matched by women reports (fear and calm). In most of the emotions SD of reported arousal values, for both genders, are similar and around 2, except for fear where is 1.5 for both genders (the lowest value).

With respect to dominance reported, where rating was from 1 for “feeling totally dominated” to 9 for “feeling strong control power”, the largest differences from the gender perspective with respect to mean value are found. Men reported higher values than mean value in anger, fear, contempt, disgust, sadness, tedium, calm and tenderness, matching better the expected values in disgust, anger, calm and tedium (high values). However, women reported higher values than the median value in surprise, attraction, and hope emotions, matching better the expected values (especially low values for fear and high values for hope and attraction). The largest difference between genders are in sadness (0.5), hope (0.6), fear (0.4), attraction (0.5) and tedium (0.6), where women report values closer to objective in all of them. With regards to SD, values are similar in all emotions for both genders and around 2.

ANOVA results show differences related to gender analysis of discrete emotions (joy, sadness, surprise, contempt, hope, fear, attraction, disgust, tenderness, anger, calm and tedium) and PAD variables (valence, arousal and dominance) (see Table 6) for every reported emotion (whatever video clip produced the label). In Table 6 it is observed how in the case of the values reported for arousal, gender differences are found when labelling the emotions of contempt (*p*-value < 0.001), hope (*p*-value = 0.01), fear (*p*-value < 0.001), attraction (*p*-value = 0.05), calm (*p*-value = 0.01) and tedium (*p*-value = 0.05). In these cases, men tend to report higher values, except in the emotion of contempt and fear, where women show a higher activation. 

Differences between women and men are also observed for self-reported valence in the emotions of joy (*p*-value < 0.001), contempt (*p*-value = 0.01), fear (*p*-value < 0.001) and calm (*p*-value < 0.001), with women tending to report higher values, except in the emotion of contempt which is reported by men. And in the case of dominance, gender differences are observed in the emotions of joy (*p*-value = 0.05), sadness (*p*-value = 0.01), hope (*p*-value = 0.01), fear (*p*-value < 0.001), attraction (*p*-value = 0.01) and tedium (*p*-value = 0.05). In these cases, men report greater control over the emotion except in the case of the emotions of attraction and hope.

The results obtained for fear emotion are especially interesting, where differences have been found regarding the gender of participants in all PAD variables (valence, arousal and dominance), being more extreme the answers from women.

### 3.2. Gender Differences in Emotion Labelling onto Affective Space Quadrants

Every quadrant in the Affective Space has been analysed with respect to the reported emotions by gender through confusion matrices. A great deal of agreement can be observed in the reported label by the participants and target emotion stated by the research team for every video clip selected. In Figure 4 the emotion labelling distribution for audiovisual stimuli aimed at emotions in quadrant Q1 is shown. Video clips corresponding to joy emotion (V01–V04) are those with a higher degree of agreement (higher than 70%) being even higher for women than for men. Video clips for surprise emotion (V06–V07) and hope (V05) are those with lower percentage of agreement, especially in the answers for video V07 where women reported the same emotion as the target in a 68% but men in only 32%, sharing the reported emotions with disgust and joy.

In Figure 5 the emotion labelling distribution for audiovisual stimuli aimed at emotions in quadrant Q2 is detailed, split by the gender of participants. In this quadrant a higher number of stimuli has been included, as UC3M4Safety database main objective is differentiating fear from the other emotions. Results presented in Figure 5 show that video clips corresponding to fear emotion (V11–V29) have the higher level of agreement for both (women and men). Especially noticeable is agreement among women, providing values above 0.9 in stimuli V15, V25 and V28. In fact, it is interesting to note that the videos V11, V28 and V27 (stimuli labelled by the research team as video clips with interest in their content related to gender-based violence) show the greatest gender differences being labelled mostly by women as “fear” while men range from “fear” to “anger” or “sadness”. For anger emotion (V08–V10), the lowest levels of agreement have been obtained, for women and men, being a bit higher for women but sharing the level of agreement with disgust emotion (quadrant Q3) for V08 and V09.

In Figure 6 the distribution of the degree of agreement by gender is shown, for the audiovisual stimuli corresponding to emotions in quadrant 3 (Q3). The results presented show women have higher levels of agreement with the target emotion than men. Video clips for sadness emotion (V34–V36) have larger differences between genders, despite the high level of agreement. Women reported values over 0.8 while values reported by men are in the range of 0.55–0.78. Video clips for disgust emotion (V31–V33) show a level of agreement over 0.60 for men and over 0.71 for women. In the case of the emotion of tedium (V30), it is the Q3 stimulus that has the least degree of agreement, especially among women.

Finally, in Figure 7 the level of agreement reported by participants, split by gender, for the audiovisual stimuli with emotions located in quadrant 4 (Q4) is detailed. The results provided show differences in emotion reporting even within men or women. The two video clips for gratitude (V37–V38) present a contrary level of agreement, higher values from men in V37 and from women in V38, this one with the lowest values in the quadrant. Video clips for calm emotion (V39–V42), also got contrary levels of agreement: videos V40 and V42 have higher level of agreement in the target emotion from men while V39 and V41 from women; although all of them with high values, over 0.67.

In order to know which of the gender differences found in the percentage of agreement for the labelling of the emotions of the UC3M4Safety database stimuli can be consider a tendency in our sample, a post-hoc study has been applied (as previously mentioned in Section 2.4—Data Analysis Section). A X^2^ test showed a dependency does exist (*p*-value = 5.373 × 10^−10^) between the discrete emotion reported and the gender of the participant.

Pearson standardized residual values are provided in Table 7, taking as reference the Z-factor (2.31). That means that residual values whose absolute value were higher than Z-factor will imply a strong dependence of emotion with gender. Through this analysis a strong relation of dependency has been found in fear and contempt emotions. Women used fear label 1.27 times more than men when classifying the emotion elicited after video clips visualization. On the other hand, men reported contempt emotion twice more than women, even for videos targeted at different emotions, although no selected video clip has this emotion as target.

## 4. Discussion

In this work, a study has been done to obtain a complete and high-quality data set of audiovisual stimuli to trigger emotions under a controlled scenario. This data set is intended for collecting further human responses (physiological and/or physical variables) that could serve in artificial intelligence-based systems aimed to distinguishing in real time and automatically an emotion. Although main purpose is fear or panic distinguishing, the finely-tuned selection of video clips, for a range of 12 emotions, together with the complete labelling system applied will serve to research and develop different artificial intelligence-based systems that require the classification of emotions felt by humans in many applications. The process of selecting and analysing a set of audiovisual stimuli for emotion elicitation, adequate for distinguishing fear emotion from the other emotions has provided four main issues that enrich the results of the research and could help to better understand the process of feeling emotions.

Firstly, regarding the question “Are emotions best labelled in the PAD space or with the discrete values?”, the emotion categorization is differently and hardly understood by people being numerical parameters of valence, arousal and dominance worse than discrete values, but not only. Emotion labelling is a complex task that is affected by many circumstances around volunteers and by their own personalities. The emotion values reported have presented expected deviations but not so expected different mean values with respect to the participants’ gender; especially for arousal parameter which is implicitly associated to positive emotions (low arousal) and negative emotions (high arousal), moving quadrants Q1 and Q3 to Q4 and Q2, respectively. According to other authors [62], emotions are not single spots in the PAD space but rather clouds of points that can cover wide areas, even overlapping other emotion-clouds. Reporting also the intensity of the emotion felt can contribute to discern labels in those frontiers and to train correctly a possible artificial intelligence system. Very few audiovisual stimuli presented in this research to the volunteers are arousing emotions in some extremes of these clouds, such as video clip V06 or V31: but for these cases, we assure a better matching between discrete emotions classification and PAD space classification is possible when the intensity of the emotion elicited were reported.

Furthermore, and from the gender point of view (and related also to the third question as it is imbricated in all the presented research), the results in Table 5 and Table 6 show the tendency that men and women have a different arousal when labelling fear or contempt emotion, being higher always on women. These gender differences that are observed in the results have their origin in the differential socialization of men and women [53,63,64], and their consequent differential education around emotions [65]. Studies in this area have found, for example, that women are more precise in the recognition of non-verbal communication or emotional prosody [47].

Secondly, for the question “Is there a high level of agreement in the reported emotions?” the obtained results, as expected, showed no complete agreement with the target emotion in any video clip. The reported discrete emotions versus target emotions have an agreement above 5% in 50 video clips and in 3 between 40–50% of agreement. This agreement is higher in *extreme* emotions such as joy, fear and sadness (see Figure 4 and Figure 7).

These differences between participant labelling (reported) and expected labelling (target), are present in other studies as well [55], may be due to a light misunderstanding with the labelling terms, as they are not common in spoken language. In future research, the values reported by the participants, especially in the case of arousal, could be contrasted with the physiological signals of the volunteer to study the levels of body activation with electroencephalogram (EEG), heart rate (HR) or skin conductivity [66].

The high level of agreement in negative emotions, mainly for fear emotion, can be related with the high identification that violence narratives are arousing in the public, due to the fact that fictional audiovisual account is looking for universal references, while presenting local difficulties and stories strongly rooted in collective memory, or the recreation of contemporary experiences close to producers and audience. In this sense, Nussbaum [67] argues emotions are not affective senseless impulses, but smart responses according to life events and personal goals and values. Therefore, the gender differences found between men and women in quadrant Q2 can be a starting point for further research of wearable systems for vulnerable people protection.

A large proportion of the participating women experience more intense and negative emotions than men with these video clips. These data may be related to previous experiences in women; according to United Nations figures, one in three women in the world has suffered physical or sexual violence, mainly from a partner [68]. Gender-based violence is a very frequent traumatic experience, for example: in Spain, half of the women declared that they had suffered gender-based violence in 2019 [69], or also, and related to these experiences, there may be a stronger identification with the characters, with women empathizing more with these audiovisual stimuli due to the activation of autobiographical memory [26,41]. Besides, there have been three video clips, formerly labelled by the research team as “fear with gender-based violence interest” that have been labelled by survey participants as anger by majority.

Recently, Israelashvili et al. [70] have pointed out that empathy and personal distress have the opposite impact in the emotion recognition in the other; personal distress affects negatively to emotion observation while empathy worrying has a positive influence. This assertion can explain the women identification in video clips V11, V28 and V27 (stimuli labelled by the research team as video clips with interest in their content related to gender-based violence), where some fear and anxiety scenes are displayed showing the female main character, and her son, under life risk due to the constant harassment by the aggressor. This fear to the harassment or aggression presents a strong gender component; as, for example, UN estimates the 35% of worldwide women has suffered physical and/or sexual violence from someone different from their couples (this figure is not including sexual harassment) along their lives [68], or in the European Union, one out of ten women declares cyber-harassment from age 15 [71].

Furthermore, the violence against women presents a prominent emotional appearance in fear, although victims try to hide it [72]. Also, it should be taken into account that the fear perceived by women victims of gender-based violence is closely related to their traumatic experiences and their duration [39], which means that factors such as the development of an anticipatory risk perception must be taken into account in the development of this research [54,73,74] that can affect both the physiological measurement of emotional responses, reported emotions or the evocation of the objective emotion—in this case ‘fear’—through a stimulus in a laboratory. However, although fear is one of the most difficult emotions to elicit in the laboratory [55], in our results it seems that the UC3M4Safety database stimuli manage to evoke this emotion better.

On the other hand, the audiovisual stimuli customization appears as one of the most important elements in this research. In the same sense as there are cultural differences between Asian and European persons in emotion elicitation [23], the presence of female main characters in video clips, chosen for fear emotion elicitation, is showing the gender differences and how gender socialization affects when facing the same situation: women label fear while men range from “fear” to “anger” or “sadness”, showing in the case of anger how the protector role with women is reproduced instead of empathizing with the victim.

Thirdly, for the question “Are emotional responses different for women and men?” the obtained results confirm gender differences in emotion elicitation with the audiovisual stimuli used in this research for this sample, not only for PAD values but also for the discrete emotion and, specially, for the negative emotions like fear and contempt. These results complement previous researches from other authors, such as the study from Chen et al. [48] that proves women are more sensitive to emotional expressions in interpersonal interactions. Analysing the results presented we observe “fear” discrete emotion presents differences in its elicitation with respect to gender. Women label differently audiovisual stimuli in the discrete emotion and also in the PAD space, mainly in valence and arousal parameters, being more extreme the reported values from them. This result is very interesting for UC3M4Safety database with regards to protect women. Besides, there are other emotions that are labelled differently, in the PAD space, by men and women, such as sadness, contempt, hope, attraction, calm and tedium, with the highest difference in hope. On the other hand, there are slight differences in joy, surprise, disgust, tenderness, and anger. With regards to the discrete emotion, men tend to label anger some of the stimuli labelled as fear by women and by researchers.

Fourth, the final question “Can we select a representative and small set of audiovisual stimuli to elicit emotions adequately?” has a positive answer, as a carefully selected set of video clips has been obtained, after applying criteria required by machine learning algorithms and statistical analysis. Particularizing the main contribution of the research, first paragraph of this Section, this data set can be used for an automatic system to protect vulnerable people, detecting “fear” emotion that can be associated with risky situations, as database for training the artificial intelligent detecting algorithm. This database has been called UC3M4Safety database and is intended for generating the secondary inputs required to train the artificial intelligence algorithms of Bindi system, as well as to be shared with the scientific community.

## 5. Conclusions

In this paper a study on the identification of emotions elicited after the visualization of audiovisual stimuli has been presented. A statistical study of gender differences in emotional response was carried out on 1332 participants (811 women and 521 men). The research study has produced a dataset of 42 audiovisual stimuli (UC3M4Safety database) that triggers a complete range of emotions, with a high level of agreement and with a discrete emotional categorization, as well as quantitative categorization in the Pleasure-Arousal-Dominance (PAD) Affective Space. This database is adequate for the machine learning algorithms contained in these automatic systems. Furthermore, the differences with respect to the gender of the participants, the method of emotion categorization and the level of agreement have been analysed and discussed.

The selection of the best stimuli in terms of agreement, visualizations and relevance allow the research team to produce an audiovisual data base useful to train artificial intelligent algorithms complying with the requirements of machine learning discipline. These algorithms are intended for automatic systems that protect vulnerable groups of population, women in the case of gender-based violence but not only, detecting clearly and rapidly risky situations. An example of one of these systems is Bindi that is aimed to detect and prevent violent aggressions on women; but the database can be used for many other protection purposes.

Obtained results show the emotions reported, discrete values (but also in the PAD space), by both genders are similar in positive emotions, while for negative ones (especially fear and contempt) these reports are more different. Autobiographical memory is influencing the fear perception, especially with those video clips identified as interesting for gender-based violence.

The gender-based violence is difficult to be labelled and, even, identification with the characters in the video clip is affecting to a great extent. Women tend to label mostly “fear” while men can label them as “anger” or “sadness”. The better labelling of fear from the women benefits our main objective of developing an automatic system to protect them for violent or sexual aggressions. 

Taking into account the obtained results, the gender variable should be considered both in the database stimulus selection stage and in the machine-learning algorithm training stage. Even though for some of the emotions there are not great differences, for others (especially fear and hope), gender differences in the reported emotion have to be considered in order to improve emotion detection.

This is especially important in this work since the main objective is the detection of fear-triggered situations in women victims of gender violence. In this case, gender must be taken into account since the emotion perceived for fear video stimuli shows differences in both genders. Even more for the particular case of stimuli reproducing situations of gender violence, where a great difference is found when labelling between women and men (fear versus anger and sad).

As future works, the authors aim to complete this UC3M4Safety database with physiological and physical measurements during emotion triggering, customizing different versions of it for the general population and for post-traumatic stress subsets.

## Figures and Tables

**Figure 1 ijerph-17-08534-f001:**
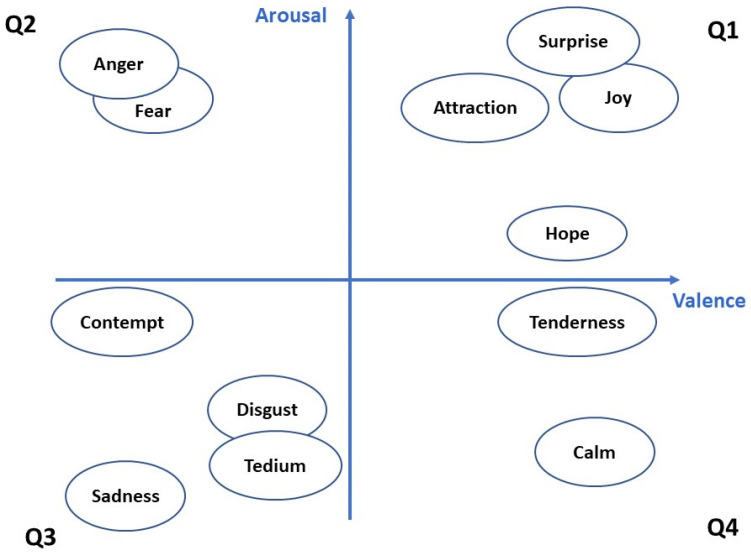
Discrete emotions draft mapping in a 2-dimensional space for arousal and valence.

**Figure 2 ijerph-17-08534-f002:**
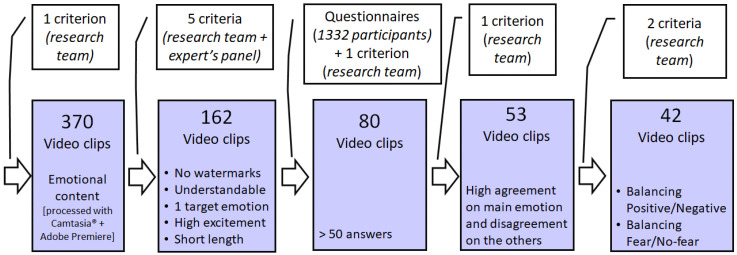
Processing of audiovisual materials in the process of creation a balanced set of video clips for fear/no-fear detection system.

**Figure 3 ijerph-17-08534-f003:**
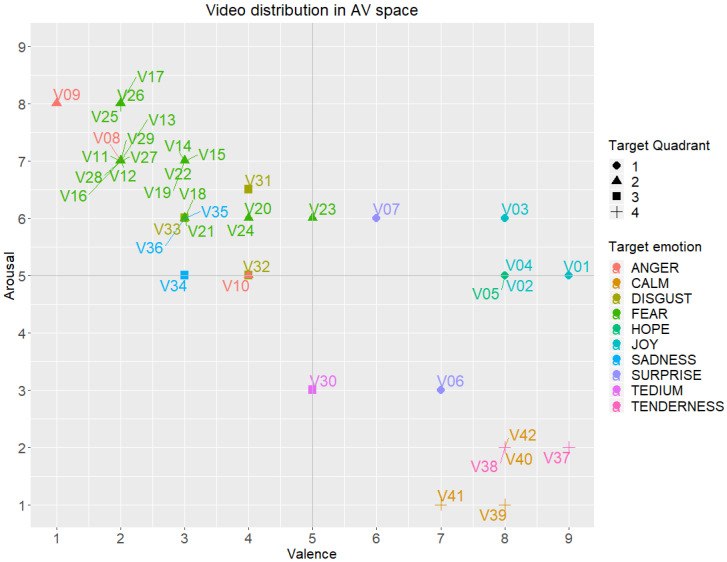
Distribution of the reported emotions of the UC3M4Safety database videos in the AV space by target emotion and quadrant.

**Figure 4 ijerph-17-08534-f004:**
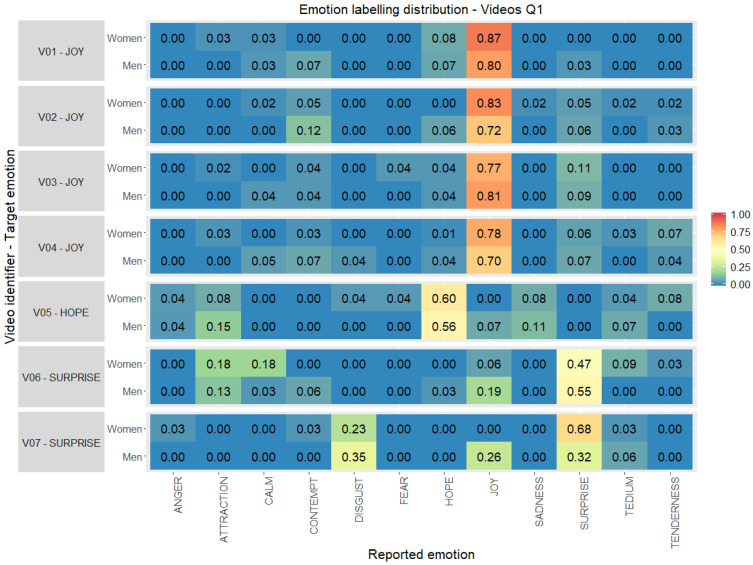
Emotion labelling distribution (0.00–1.00) between the emotions reported by the volunteers with respect to the original target per video clip selected, gender and quadrant 1 (percentage computed considering individually women and men totals per stimuli).

**Figure 5 ijerph-17-08534-f005:**
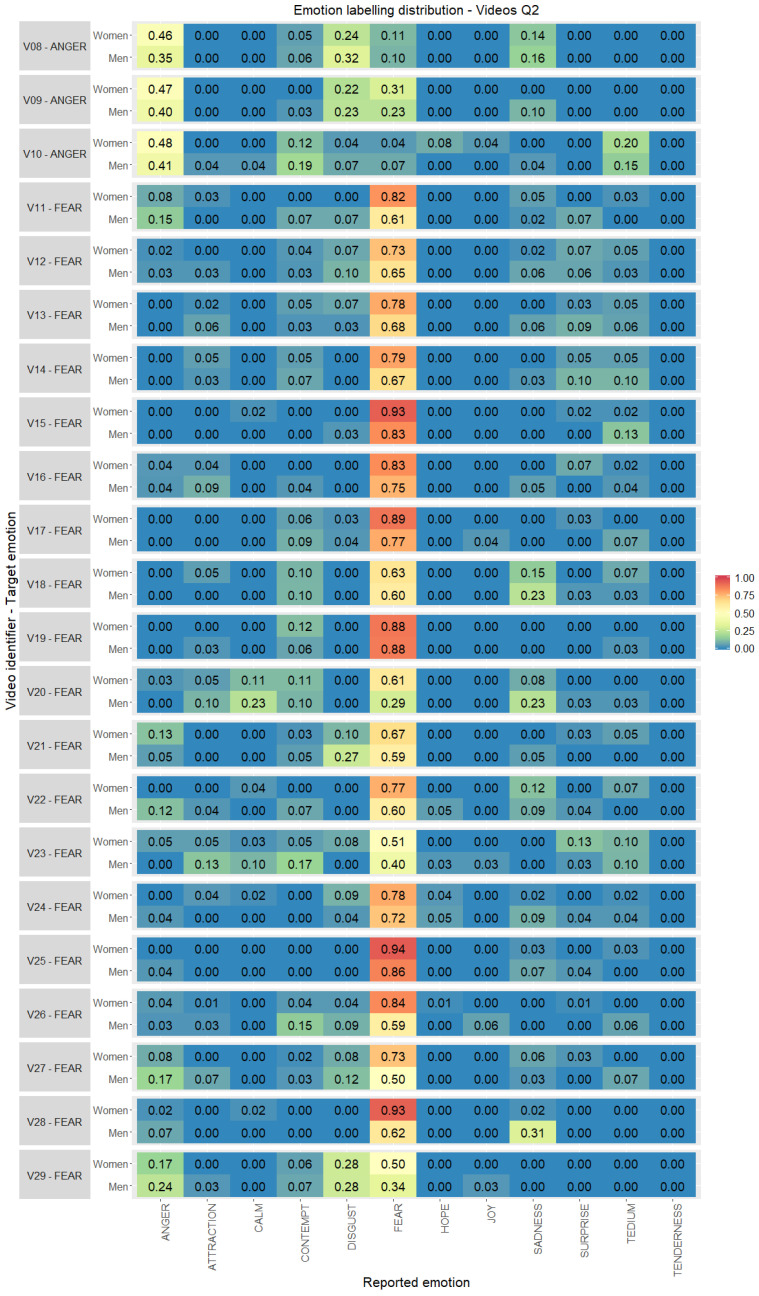
Emotion labelling distribution (0.00–1.00) between the emotions reported by the volunteers with respect to the original target per video clip selected, gender and quadrant 2 (percentage computed considering individually women and men totals per stimuli).

**Figure 6 ijerph-17-08534-f006:**
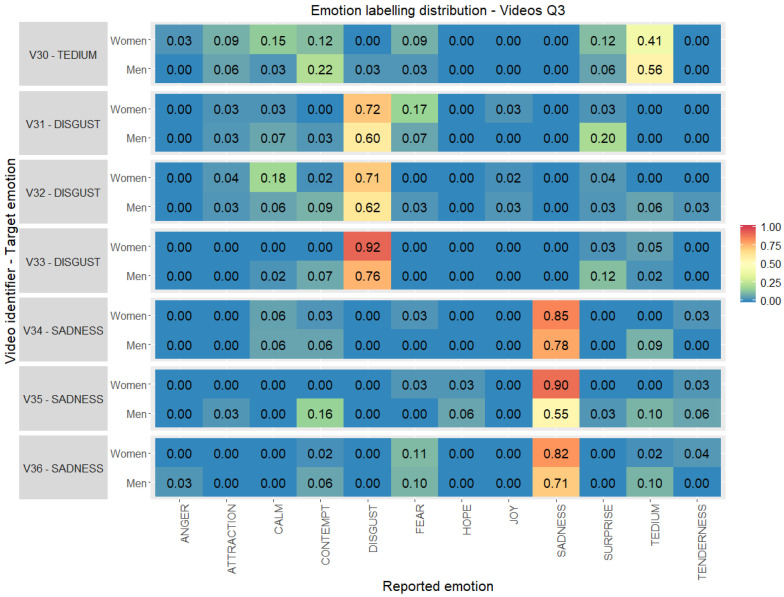
Emotion labelling distribution (0.00–1.00) between the emotions reported by the volunteers with respect to the original target per video clip selected, gender and quadrant 3 (percentage computed considering individually women and men totals per stimuli).

**Figure 7 ijerph-17-08534-f007:**
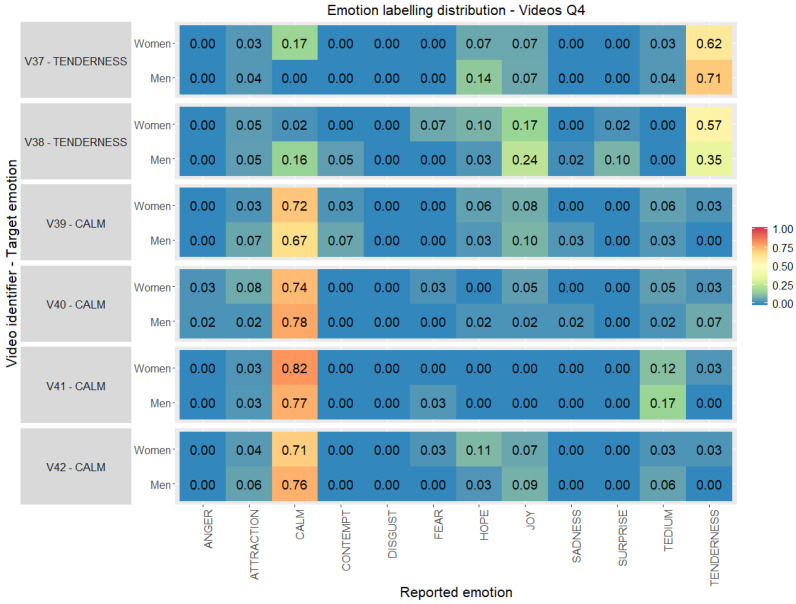
Emotion labelling distribution (0.00–1.00) between the emotions reported by the volunteers with respect to the original target per video clip selected, gender and quadrant 4 (percentage computed considering individually women and men totals per stimuli).

**Table 1 ijerph-17-08534-t001:** Emotion classification considered in this work for the search of audiovisual stimuli and their subsequent labelling.

Positive Emotions	Negative Emotions
**Joy**	**Sadness**
**Surprise**, amusement	**Contempt**, indifference
**Hope**, trust, pride, achievement	**Fear**
**Attraction**, desire, interest, admiration	**Disgust**, aversion, revulsion
**Tenderness**, gratitude, contentment, satisfaction	**Anger**, rage, fury
**Calm**	**Tedium**, boredom

**Table 2 ijerph-17-08534-t002:** Number of video clips per target emotion surveyed and analysed.

Emotion	No. Surveyed Video Clips: 162 (No. Analysed for DB (80))
Joy	13 (5)
Sadness	12 (6)
Surprise	18 (4)
Contempt	10 (8)
Hope	8 (6)
Fear	40 (21)
Attraction	6 (6)
Disgust	8 (4)
Tenderness	12 (10)
Anger	12 (1)
Calm	10 (6)
Tedium	13 (3)

**Table 3 ijerph-17-08534-t003:** Distribution of videoclips selected in UC3M4Safety Database according to AV space.

Target Emotion	Quadrant in AV Space	N Videos
Joy	Q1	4
Sadness	Q3	3
Surprise	Q1	2
Contempt	Q3	0
Hope	Q1	1
Fear	Q2	19
Attraction	Q1	0
Disgust	Q3	3
Tenderness	Q4	2
Anger	Q2	3
Calm	Q4	4
Tedium	Q3	1

**Table 4 ijerph-17-08534-t004:** Data from the sample of participants by selected videos in the UC3M4Safety Database.

N Videos	Average Votes	SD Votes	Max–Min	Mean (Women)	SD (Women)	Max–Min (Women)	Mean (Men)	SD (Men)	Max–Min (Men)
42	83,881	22,106	129–52	46,309	14,195	75–25	37,429	11,236	62–27

**Table 5 ijerph-17-08534-t005:** Self-report responses per emotion reported category, mean (standard deviation).

Emotion Category	Gender	Valence Mean (SD)	Arousal Mean (SD)	Dominance Mean (SD)
Joy	General	8.1 (1.1)	4.4 (2.5)	7.0 (2.0)
	Women	8.2 (1.1)	4.4 (2.5)	7.1 (2.1)
	Men	7.9 (1.1)	4.3 (2.4)	6.8 (2.0)
Sadness	General	3.1 (1.7)	5.6 (2.0)	5.5 (2.2)
	Women	3.0 (1.8)	5.8 (2.0)	5.3 (2.3)
	Men	3.2 (1.7)	5.4 (2.0)	5.8 (2.0)
Surprise	General	6.3 (1.7)	4.8 (2.0)	6.6 (1.8)
	Women	6.3 (1.7)	4.8 (2.0)	6.7 (1.9)
	Men	6.2 (1.6)	4.8 (2.0)	6.6 (1.8)
Contempt	General	4.5 (1.4)	4.0 (2.1)	7.0 (1.9)
	Women	4.3 (1.4)	4.4 (2.2)	6.9 (1.9)
	Men	4.7 (1.4)	3.7 (2.0)	7.1 (1.9)
Hope	General	7.5 (1.6)	4.2 (2.3)	7.1 (1.8)
	Women	7.6 (1.6)	3.9 (2.3)	7.4 (1.8)
	Men	7.4 (1.6)	4.6 (2.2)	6.8 (1.8)
Fear	General	2.6 (1.6)	7.2 (1.5)	4.1 (2.2)
	Women	2.4 (1.6)	7.4 (1.5)	4.0 (2.3)
	Men	2.9 (1.5)	6.9 (1.5)	4.4 (2.1)
Attraction	General	6.3 (1.7)	4.5 (2.2)	7.1 (1.8)
	Women	*6.5 (1.7)*	*4.3 (2.3)*	7.3 (1.7)
	Men	6.2 (1.7)	4.8 (2.0)	6.8 (1.8)
Disgust	General	3.1 (1.8)	6.0 (2.0)	5.3 (2.4)
	Women	3.2 (1.8)	6.0 (2.0)	5.2 (2.5)
	Men	3.1 (1.8)	6.0 (1.9)	5.4 (2.3)
Tenderness	General	7.9 (1.3)	3.5 (2.3)	6.9 (2.3)
	Women	8.0 (1.2)	3.6 (2.3)	6.8 (2.4)
	Men	7.7 (1.4)	3.4 (2.3)	7.0 (2.1)
Anger	General	2.4 (1.6)	6.6 (1.7)	5.1 (2.2)
	Women	2.4 (1.6)	6.7 (1.7)	5.1 (2.3)
	Men	2.5 (1.5)	6.6 (1.6)	5.2 (2.1)
Calm	General	7.1 (1.6)	2.4 (1.8)	7.5 (1.7)
	Women	7.4 (1.6)	2.2 (1.6)	7.4 (1.9)
	Men	6.9 (1.6)	2.6 (1.9)	7.7 (1.6)
Tedium	General	4.7 (1.3)	3.5 (2.1)	6.8 (2.1)
	Women	4.8 (1.5)	3.7 (2.2)	6.5 (2.1)
	Men	4.7 (1.0)	3.3 (1.9)	7.1 (2.0)

**Table 6 ijerph-17-08534-t006:** *p*-Values results from ANOVA for arousal, valence and dominance reported by the volunteers according to gender.

Emotion	Arousal	Valence	Dominance
**Joy**	0.834	0.00027 ***	0.0123 *
**Sadness**	0.0604	0.236	0.00621 **
**Surprise**	0.786	0.669	0.743
**Contempt**	0.00047 ***	0.00256 **	0.427
**Hope**	0.00694 **	0.169	0.00523 **
**Fear**	4.25 × 10^−9^ ***	1.42 × 10^−7^ ***	0.000956 ***
**Attraction**	0.0314 *	0.0946	0.00463 **
**Disgust**	0.722	0.922	0.471
**Tenderness**	0.494	0.0824	0.374
**Anger**	0.449	0.28	0.708
**Calm**	0.0088 **	0.000137 ***	0.0684
**Tedium**	0.0314 *	0.782	0.0113 *

NOTE: Significance codes: ‘***’ 0.001 ‘**’ 0.01 ‘*’ 0.05 ‘.’ 0.1 ‘ ’ 1.

**Table 7 ijerph-17-08534-t007:** Pearson standardized residual values (Z-factor) between the discrete emotion reported and gender.

	**Joy**	**Sadness**	**Surprise**	**Contempt**	**Hope**	**Fear**
Women	−1.5559	−0.4584	−1.7373	−5.2307 **	−0.7867	6.0483 **
Men	1.5559	0.4584	1.7373	5.2307 **	0.7867	−6.0483 **
	**Attraction**	**Disgust**	**Tenderness**	**Anger**	**Calm**	**Tedium**
Women	−1.2545	1.8722	−0.1990	1.1330	−0.9483	−0.6388
Men	1.2545	−1.8722	0.1990	−1.1330	0.9483	0.6388

NOTE: Significance codes: ‘**’ < (−2.31) and > (2.31).

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
