# Peer review of "Emotion Elicitation Under Audiovisual Stimuli Reception: Should Artificial Intelligence Consider the Gender Perspective?"

_ijerph, 2020, doi:10.3390/ijerph17228534_

Round 1

Reviewer 1 Report

This is an interesting research that can contribute to some future research fields relating human emotions. For example, the methods and results of this research can be applied to train artificial intelligence algorithms that can identify human emotion using audio-visual data.   

However, I can find some problems in the several parts of this draft. I hope the authors can improve the final draft considering the following recommendations.

  1. Title

Please re-consider the title of this research. I think the main subject of this draft is “the method of identifying human emotions in the audio-visual materials and its implication”. The gender difference is only a part of the implication of the results. As I mentioned this research can contribute to the future research of AI or machine learning. Therefore, I believe the authors should focus on the method and future use of your research.

  1. Implication or discussion

When discussing the gender difference of some emotions, in particular, fear, the authors should add some previous research results in the field of risk perception. In addition, it is better to refer the previous literature about disaster and emotions of disaster victims.

One more important issue in discussion and result section is to stress the importance of this research in future research. The authors should stress that the methods and results of this research can be applied to train artificial intelligence algorithms and many other future research and industry uses.

Author Response

Muchas gracias por sus comentarios, han sido de gran ayuda para mejorar la presentación de los resultados y su alcance. 

  1. En cuanto al título, hemos considerado tu propuesta y la hemos cambiado a la siguiente: 

Elicitación de emociones bajo la recepción de estímulos audiovisuales: ¿Debería la inteligencia artificial considerar la perspectiva de género? 

más relacionado con la implicación de utilizar estímulos audiovisuales para desencadenar emociones, como sugieres. 

  1. We have also included references from previous literature about disaster and emotions of disaster victims. In this regard you can see the changes in the introduction (now split into 3 sections), where the following paragraph has been incorporated (section 1.3): 

“One of the most studied cases is the emotional response of people suffering from post-traumatic stress as a result of a traumatic experience such as war [32–35], terrorist attack [36] or natural disaster [37,38]. Fears associated with traumatic experiences are not common to all people [39] and change their perception of risk”. 

In addition, we have also referred to this essential point about the perception of risk in the hypothesis: 

“Our initial hypothesis is that men and women have different emotional responses, due to gender socialization [50]. Differences in emotional response that may be increased for women who have experienced gender-based violence because post-traumatic stress and risk perception is analogous to the aftermath of war victims [51]”

and in the discussion: 

“Also, it should be taken into account that the fear perceived by women victims of gender-based violence is closely related to their traumatic experiences and their duration [39], which means that factors such as the development of an anticipatory risk perception must be taken into account in the development of this research [51,71,72] that can affect both the physiological measurement of emotional responses, reported emotions or the evocation of the objective emotion - in this case 'fear' - through a stimulus in a laboratory”.

  1. The practical and theoretical contributions of this research have been included throughout the text (from the abstract, introduction, discussion and conclusions). Greater emphasis has also been placed on the importance of this study in artificial intelligence-based systems, not only in the discussion and conclusions, but also in the introduction

“The identification of what emotions are triggered by different sourced stimuli can be applied to automatic systems to help, relieve or protect some groups of population. Both elements, stimuli and labelled emotions are the primary inputs for an artificial intelligence-based system; that, of course, needs from secondary inputs, other types of human responses (the more unconscious the better) such as physiological variables, voice, face expressions, etc. to train the algorithms that could classify, in real time and automatically, the emotion experienced by the person. In this sense, Bindi system is proposed by the UC3M4Safety team to detect and prevent violent aggressions against women, by detecting fear or panic emotions, through the voice and the physiological responses [17–19]. However, specific databases are required to create these artificial intelligence-based algorithms in detecting fear or panic emotions and considering the users will be women. The selection of the best stimuli allows us to train and test these automatic systems in an efficient and accurate way”. 

In the discussion section, a summary is included at the beginning, substituting the former sentence:

“In this work, a research has been done to obtain a complete and high-quality data set of audiovisual stimuli to trigger emotions under a controlled scenario. This data set is intended for collecting further human responses (physiological and/or physical variables) that could serve in artificial intelligence-based systems aimed at distinguishing in real time and automatically an emotion. Although main purpose is fear or panic distinguishing, the finely-tuned selection of video clips, for a range of 12 emotions, together with the complete labelling system applied will serve to research and develop different artificial intelligence-based systems that require the classification of emotions felt by humans in many applications. The process of selecting and analysing a set of audiovisual stimuli for emotion elicitation, adequate for distinguishing fear emotion from the other emotions has provided four main issues that enrich the results of the research and could help to better understand the process of feeling emotions”

Also in the last paragraph, w.r.t. the final question (fourth) answered:

“Particularizing the main contribution of the research, first paragraph of this section, this data  set can be used for an automatic system to protect vulnerable people, detecting “fear” emotion that can be associated with risky situations, as a database for training the artificial intelligent detecting algorithm. This database has been called UC3M4Safety database and is intended for generating the secondary inputs required to train the artificial intelligence algorithms of Bindi system, as well as to be shared with the scientific community”. 

And, finally, in the conclusions section: “

“Teniendo en cuenta los resultados obtenidos, la variable de género debe considerarse tanto en la etapa de selección de estímulos de la base de datos como en la etapa de entrenamiento del algoritmo de aprendizaje automático. Aunque para algunas de las emociones no hay grandes diferencias, para otras (especialmente el miedo y la esperanza), las diferencias de género en la emoción reportada deben ser consideradas para mejorar la detección de emociones. Esto es especialmente importante en este trabajo ya que el principal objetivo es la detección de situaciones de miedo en mujeres víctimas de violencia de género. En este caso se debe tener en cuenta el género ya que la emoción percibida por los estímulos del video de miedo muestra diferencias en ambos sexos. Más aún para el caso particular de estímulos que reproducen situaciones de violencia de género, donde se encuentra una gran diferencia a la hora de etiquetar entre mujeres y hombres (miedo versus enfado y tristeza). Como trabajos futuros, los autores pretendían completar esta base de datos UC3M4Safety con mediciones fisiológicas y físicas durante la activación de emociones, personalizando diferentes versiones de la misma para la población general y para subconjuntos de estrés postraumático ”..

Reviewer 2 Report

The paper is very interesting and uses the results of extensive research. Despite this, it is necessary to clarify some of the assumptions adopted by the authors, in particular regarding the research sample.

The authors made a common mistake in assuming that the snowball sample is a random sample. In fact, the selection of units to be tested in this case should be considered purposeful (non-random sample). If the sample is a deliberate sample and it cannot be considered even as a quasi-random, statistical inference methods should not be used to describe the results of the study, or at least all the time it should be emphasized when describing them that it is only a tendency appropriate only for this study.

We’re dealing with random sampling whenever the following conditions are met:

  1. Every element in our population has a nonzero probability of being selected as part of the sample.
  2. We have accurate knowledge of this probability, known as the inclusion probability, for each element in the sampling frame.

Therefore, I recommend much greater caution, especially when examining the statistical significance of parameters or their relationships.

Author Response

Thank you very much for your comments, they have been very useful to improve the presentation of the results and the methodology. As you indicated, we cannot claim the randomness of the sample since we did not have any database that would allow the total Spanish population to have the same possibilities to participate. For this reason, in section 2.3. Sample, we have eliminated "randomly". 

We have also added a sentence, detailing the analysis performed, in section 2.4. Data Analysis: 

“In addition, to observe tendencies of dependency related to gender analysis, and after checking the homogeneity of the variance (Levene’s test) and normality (QQ-plot) assumptions … ”

to the previously existing paragraph to ensure the validity of the results of statistical studies applied to this sample. 

In the Results section, but also in Discussion and Conclusions, as the sample can not be considered random and to avoid generalization and statistical inference, the sentences including statistical significance or significant relationships have been changed for words like "differences" or “relevant tendencies”. 

Reviewer 3 Report

The main purpose of the presented in the paper was to select audiovisual stimuli that elicit different emotional responses (especially fear) in women. The selected audiovisual stimuli would then be used to train machine learning algorithms to recognise different situations and their potential emotional interpretation by vulnerable people groups. Various other questions were raised and answered by the research, such as differences in emotional responses and interpretations by genders. Audiovisual stimuli were selected and presented to 1332 participants to view and report their emotions elicited. Data was then used to narrow down the selection of stimuli for machine learning, as well as to address the research questions that were raised tangentially to the main purpose of the report.

I have several comments to improve the manuscript:

  1. First, the writing of the manuscript is unclear and difficult to follow. As a whole, every segment of the paper requires a deep review of the grammar and formatting.
  2. The introduction, though in-depth and informative, was rife with grammatical errors (e.g., line 6 “basic emotion theory researches”, line 33 “the identification of what emotions are provoking different sourced stimuli” – it is important that directionality shown implied is correct). Beyond grammatical errors, many lines require rephrasing as the purpose of the sentences are not conveyed clearly (e.g., line 52 and 58).

  3. There is a lack of theoretical discussion in the introduction related to gender differences in emotional interpretation of situational stimuli. The authors should include a portion in the introduction that looks into research on gendered differences in emotional interpretation of situational stimuli as this was an area of research interest in the paper.
  4. The introduction of the paper could be made clearer with sub-headers used to identify different portions of the research phenomenon introduced.

  5. The research questions listed down in the introduction of the paper do not tie in with the order in which they are referenced in the results section. The authors might want to consider rearranging these questions either in the introduction or the results section, keeping in mind to apply the same changes in the discussion portion.

  6. For the description of the sample, a clearer explanation as to why the original sample of 1520 participants was reduced to 1332 participants should be given.

  7. Figure 2 had a spelling error in one of the boxes. Table 2 requires further specification of ‘Nb’ as it is unclear to readers.

  8. In the paragraph just after table 2, the reported threshold of 40% requires substantiating. It is unclear why the authors chose this specific threshold. More rationale is necessary

  9. In Table 6, the decimal point for *** is missing. The p value should not be zero.
  10. The internal consistency of the measures should be reported
  11. The practical and theoretical contributions of the current should be elaborated further
  12. I was unable to access the supplementary materials. The paper will have important contribution to the field if the video clip materials and the emotional ratings are fully shared to the readings.

Author Response

Dear Reviewer No. 3

Thank you very much for your comments, they have been very useful to improve our paper. Below we will indicate the detailed response to each of your comments and the correction that has been made: 

  • (1) The text has been reviewed in a final reading by a native colleague.
  • (2) Corrections and improvements to the English language as indicated by reviewer number 3. 
  • (3) In the Introduction section, six references to previous studies that have found gender differences in emotion elicitation and their details have been included (see reference 42, 47, 48, 49, 50 and 51). In addition, we have also included references related to differential gender socialisation (references 52 and 53) and risk perception in victims of gender-based violence (reference 54). This topic has also been addressed in greater depth in the Discussion section (also including references 63, 64 and 65 on social issues in differential socialisation and its influence through emotions such as love; 72, 73, 74 in relation to gender-based violence and fear). 
  • (4) The Introduction section has been restructured with different subtitles (1.1. Human emotions and their categorization, 1.2. Triggering Emotions and its usefulness, 1.3. Research objectives). 
  • (5) The order of the Research Questions in all the sections has been reorganised to match with the order in results presentation and discussion. 
  • (6) A clearer explanation of the sample has been included. The units of analysis are the 42 video clips of the UC3M4Safety database. Although the survey was completed by 1520 participants, there were some clips that didn’t fulfill the requirements and they were discarded. Therefore, only the answers from 1332 participants were included in the presented study. In the poll, each participant completed one questionnaire with 5 or 6 videos, if all those videos were discarded then this volunteer was discarded. On the other hand, if at least one of those videos was considered good for analysis, that participant was kept for the study. To avoid confusion the initial number of participants has been eliminated and only the number of participants included in the study is included. Section 2.3 Sample:

“The analysis units were the 42 videos of audiovisual stimuli selected for the UC3M4Safety database. A statistical study of gender differences in emotional response was carried out for the 1332 participants (811 women and 521 men) corresponding to the selected videos. The participants were 18-78 years old (mean age 38.27, SD = 14.47), and all were Spanish speakers. (...)” 

  • (7) The abbreviation of number “Nb.” has been substituted by No. in table 2.
  • (7) Two spelling errors in the figure 2 have been corrected: “criterion” in Step1 and Saxon genitive in Step 2.
  • (8) The percentage of agreement criteria taken into account for the selection has been clarified. 

For the final selection of the audiovisual stimuli (step 4, see Figure 2), two conditions were set: the first one looked for the highest percentages of agreement among the participants, meaning at least 50% of the volunteers considering both genders together or at least 50% of one gender individually, who visualized each stimulus labelled it with the same discrete emotion. At the same, the second imposed condition checked the uniqueness of that label, by ensuring that all the other possible emotions only reached as maximum a 30% agreement. Finally, due to the complexity shown by the anger emotion, which even worsened when the clip was related to gender-based violence, the research team decided to keep the stimuli labelled with this target emotion which reached at least 40% of agreement to be able to study the subsequent label evolution in future experiments with gender-based violence victims. Only in four videos (V08, V09, V10, V29) the objective emotion labelled by the researchers was changed by the majority of the volunteers.

  • (9) The error in table 6 has been corrected, removing the number 0 that was not necessary. 
  • (10) The internal consistency of the measures is reported Before applying ANOVA test the internal consistency of the data, the homogeneity of the variance (using a Levene’s test) and normality (using QQ-plot) assumptions were studied and checked.
  • (11) The practical and theoretical contributions of this research have been reinforced throughout the text (from the abstract, introduction, discussion and conclusions). From abstract:

“(...) The selection of the best stimuli allows to train these artificial intelligence-based systems in a more efficient and precise manner in order to discern different risky situations, characterized either by panic or fear emotions, in a clear and accurate way. The presented research study has produced a dataset of audiovisual stimuli (UC3M4Safety database) that triggers a complete range of emotions, with a high level of agreement and with a discrete emotional categorization, as well as quantitative categorization in the Pleasure-Arousal-Dominance Affective Space. This database is adequate for the machine learning algorithms contained in these automatic systems. Furthermore, this work analyses the effects of gender in the emotion elicitation under audiovisual stimuli, which can help to better design the final solution. Particularly, the focus is set on emotional responses to audiovisual stimuli reproducing situations experienced by women, such as gender-based violence.(...)”

From the introduction (section 1.2):

“(...) The identification of what emotions are triggered by different sourced stimuli can be applied to automatic systems to help, relieve or protect some groups of population. Both elements, stimuli and labelled emotions are the primary inputs for an artificial intelligence-based system; that, of course, needs from secondary inputs, other types of human responses (the more unconscious the better) such as physiological variables, voice, face expressions, etc. to train the algorithms that could classify, in real time and automatically, the emotion experienced by the person. In this sense, Bindi system is proposed by the UC3M4Safety team to detect and prevent violent aggressions against women, by detecting fear or panic emotions, through the voice and the physiological responses [17–19]. However, specific databases are required to create these artificial intelligence-based algorithms in detecting fear or panic emotions and considering the users will be women. The selection of the best stimuli allows us to train and test these automatic systems in an efficient and accurate way. (...)”

From discussion:
“In this work, a research has been done to obtain a complete and high-quality data set of audiovisual stimuli to trigger emotions under a controlled scenario.This data set is intended for collecting further human responses (physiological and/or physical variables) that could serve in artificial intelligence-based systems aimed to distinguish in real time and automatically an emotion. Although the main purpose is fear or panic distinguishing, the finely-tuned selection of video clips, for a range of 12 emotions, together with the complete labelling system applied will serve to research and develop different artificial intelligence-based systems that require the classification of emotions felt by humans in many applications.

and in the fourth research question:

“Fourth, the final question “Can we select a representative and small set of audiovisual stimuli to elicit emotions adequately?” has a positive answer, as a carefully selected set of video clips has been obtained, after applying criteria required by machine learning algorithms and statistical analysis. Particularizing the main contribution of the research, first paragraph of this section, this data  set can be used for an automatic system to protect vulnerable people, detecting “fear” emotion that can be associated with risky situations, as database for training the artificial intelligent detecting algorithm. This database has been called UC3M4Safety database and is intended for generating the secondary inputs required to train the artificial intelligence algorithms of Bindi system, as well as to be shared with the scientific community.”

From conclusions: 

“Taking into account the obtained results, the gender variable should be considered both in the database stimulus selection stage and in the machine-learning algorithm training stage. Even though for some of the emotions there are not great differences, for others (especially fear and hope), gender differences in the reported emotion have to be considered in order to improve emotion detection.

This is especially important in this work since the main objective is the detection of fear-triggered situations in women victims of gender violence. In this case, gender must be taken into account since the emotion perceived for fear video stimuli shows differences in both genders. Even more for the particular case of stimuli reproducing situations of gender violence, where a great difference is found when labelling between women and men (fear versus anger and sadness).

As future works, authors aimed to complete this UC3M4Safety database with physiological and physical measurements during emotion triggering, customizing different versions of it for the general population and for post-traumatic stress subsets.” 

  • (12) The supplementary materials were added to the submission, we do not know the reasons why you have not been able to consult them. However, UC3M4Safety team and its research projects are participating in open science, so all the data will be available on the project website. 

Round 2

Reviewer 1 Report

OK. This version is good. 

Reviewer 2 Report

I would like to thank the authors for the changes made. The remark concerning the randomness of the sample was taken into account. This is a common mistake when describing a research sample. It is worth paying more attention to the description of the sample selection and the related consequences in subsequent articles. The article may be published in its current form.

Reviewer 3 Report

The authors have addressed all my previous comments. Thank you so much for all the hard work. The manuscript is currently ready for publication. Well done!